# Association Between Genetic Polymorphisms in the Prostaglandin Pathway and the Development of Patent Ductus Arteriosus in Preterm Infants

**DOI:** 10.3390/ijms26199274

**Published:** 2025-09-23

**Authors:** Marcin Minta, Grażyna Kurzawińska, Zuzanna-Banach Minta, Agnieszka Seremak Mrozikiewicz, Dawid Szpecht

**Affiliations:** Department of Neonatology, Karol Marcinkowski University of Medical Sciences in Poznan, ul. Polna 33, 60-535 Poznan, Poland; gkurzawinska@ump.edu.pl (G.K.); zuzbanach@gmail.com (Z.-B.M.); neonatologia@ump.edu.pl (A.S.M.); dawid.szpecht@poczta.fm (D.S.)

**Keywords:** patent ductus arteriosus, preterm newborns, risk factors, genetics factors

## Abstract

Patent ductus arteriosus (PDA) constitutes a significant clinical condition, frequently associated with a spectrum of complications that may profoundly compromise the health status of neonates, particularly those born preterm. Multiple predisposing factors—including prematurity, low birth weight, and respiratory insufficiency—have been consistently documented in the scientific literature. In this study, we investigated the influence of genetic polymorphisms in genes associated with the arachidonic acid–prostaglandin metabolic pathway. Specifically, we analyzed polymorphisms in genes encoding phospholipase A2 (rs10798059, rs1549637, rs4375, rs1805017, rs1051931), cyclooxygenase-1 (rs1236913), prostaglandin synthase 2 (rs13283456), and the prostaglandin E2 receptor EP4 (rs4613763). The study cohort comprised 99 preterm neonates born between 24 and 32 weeks of gestation. Genetic analyses were performed to identify polymorphisms in the aforementioned genes. Statistical evaluation demonstrated that selected polymorphic were significantly associated with an increased risk of patent ductus arteriosus development. This study represents a preliminary step toward elucidating the contribution of genetic variability to the pathogenesis of patent ductus arteriosus. Improved understanding of these molecular mechanisms may facilitate the early identification of neonates at increased risk and support the implementation of targeted monitoring and preventive strategies in this high-risk population.

## 1. Introduction

Prematurity is one of the most well-established risk factors for the development of patent ductus arteriosus (PDA) and its associated hemodynamic complications. Numerous studies have demonstrated an inverse correlation between gestational age and the incidence of PDA. Additional contributing factors include low birth weight, the need for mechanical ventilation, systemic inflammation, low Apgar scores, and perinatal asphyxia. PDA has been implicated in the pathogenesis of several severe neonatal complications, including intraventricular hemorrhage, necrotizing enterocolitis, bronchopulmonary dysplasia, and retinopathy of prematurity [1,2]. Furthermore, increased mortality has been reported among neonates diagnosed with PDA.

Identifying individuals at an increased risk of developing patent ductus arteriosus is of particular importance, especially among patients receiving care in neonatal intensive care units. While traditional risk factors remain relevant, contemporary diagnostic approaches should extend beyond the identification of neonates with clinically evident PDA. Efforts should focus on proactively recognizing preterm infants with a heightened predisposition to this condition, including those with underlying genetic susceptibility.

Among neonates, an increased incidence of patent ductus arteriosus has been observed in twins, who have long served as a model for investigating genetic determinants of disease. This association is particularly pronounced in monozygotic twins, suggesting a significant heritable component [3,4]. Moreover, studies in animal models have identified genetic factors that confer susceptibility to PDA, further supporting the role of inherited predisposition in its pathogenesis [5,6].

Among the most extensively described mechanisms regulating flow through the patent ductus arteriosus (PDA) is the postnatal increase in oxygen tension, which induces constriction of vascular smooth muscle and, consequently, cessation of flow [7]. Antagonistic to the vasoconstrictive effect of oxygen is the action of prostaglandins. Prostaglandin E (PGE) plays a central role in maintaining ductal patency in utero. Its biological activity is mediated predominantly through the EP4 receptor, which is primarily expressed in vascular smooth muscle cells. Activation of this receptor stimulates nitric oxide synthase, resulting in smooth muscle relaxation. The postnatal decline in PGE_2_ concentrations, in conjunction with elevated partial oxygen pressure, contributes to ductal closure [8,9]. The synthesis of prostaglandins from arachidonic acid is catalyzed principally by cyclooxygenase-1, with the peroxidase site serving as the second catalytic domain [10].

The genes analyzed in this study are located on distinct chromosomes and participate in the arachidonic acid–prostaglandin metabolic pathway. The *PTGS1* gene, situated on chromosome 9q32–q33.3, encodes cyclooxygenase-1 (COX-1), an enzyme that catalyzes the conversion of arachidonic acid to prostaglandin H_2_. *PTGES2*, located on chromosome 9q34.11, encodes prostaglandin E synthase 2, which mediates the subsequent conversion of prostaglandin H_2_ to prostaglandin E_2_. The *PTGER4* gene, mapped to chromosome 5p13.1, encodes one of the prostaglandin E_2_ receptors (EP4), which is responsible for transducing its biological effects at the cellular level.

Several genes encoding phospholipase A_2_ isoforms also contribute to this metabolic pathway. *PLA2G4A*, located on chromosome 1q31.1, and *PLA2G4C*, situated on chromosome 19q13.3, encode cytosolic phospholipase A_2_ enzymes. *PLA2G6*, mapped to chromosome 22q13.1, and *PLA2G7*, located on chromosome 6p21.2, represent additional isoforms with distinct regulatory functions. Phospholipase A_2_ enzymes play a pivotal role in the hydrolysis of ester bonds within membrane phospholipids, facilitating the release of arachidonic acid and other polyunsaturated fatty acids, which serve as key precursors for eicosanoid biosynthesis, including prostaglandins.

The pharmacological management of patent ductus arteriosus includes the use of cyclooxygenase inhibitors, such as ibuprofen and indomethacin, as well as the peroxidase inhibitor paracetamol, alongside the option of surgical ligation [11]. None of these agents is devoid of systemic effects, particularly in the context of the physiologic immaturity of preterm neonates. Clinical evidence indicates notable interindividual variability in response to pharmacologic treatment. This heterogeneity may be partially explained by genetic polymorphisms within genes encoding enzymatic catalytic domains and prostaglandin receptors.

It is anticipated that elucidating the genetic basis of this variability and its clinical implications will enable more precise selection of pharmacological interventions and optimize therapeutic outcomes.

## 2. Results

The study group comprised 99 preterm infants born between 27 and 31 weeks of gestation, including 45 females (45.45%) and 54 males (54.55%). Patent ductus arteriosus (PDA) was diagnosed in 36 neonates, of whom 21 met the criteria for hemodynamically significant PDA (HsPDA). Pharmacological treatment was administered in 22 cases (22.22%), using either paracetamol or ibuprofen.

A statistically significant association was observed between the need for mechanical ventilation and the diagnosis of PDA, with affected infants requiring ventilatory support more frequently.

Necrotizing enterocolitis (NEC) was diagnosed in 17 neonates (17.17%), intraventricular hemorrhage (IVH) in 39 neonates (39.39%), bronchopulmonary dysplasia (BPD) in 52 neonates (52.53%), and retinopathy of prematurity (ROP) in 46 neonates (46.46%). In the analyzed cohort, a statistically significant association was found between the occurrence of PDA and the presence of NEC, ROP, and BPD.

The characteristics of the study population are summarized in Table 1. The distribution of individual single nucleotide polymorphisms (SNPs) is presented in Table 2.

Analysis of the studied polymorphisms in relation to their potential role in promoting delayed closure of the ductus arteriosus (PDA) revealed an increased frequency of PDA among carriers of the rs1051931 polymorphism.

No statistically significant associations were observed for the remaining polymorphisms of the studied genes. Detailed results are presented in Table 3.

A tendency toward delayed ductal closure was observed in neonates carrying the rs1051931 polymorphism, although the association did not achieve statistical significance (*p* = 0.099).

For the remaining polymorphisms, no significant effect on the timing of ductus arteriosus closure was identified.

Assessment of the impact of individual polymorphisms on the occurrence of hemodynamically significant PDA (HsPDA) did not reveal any statistically significant associations.

None of the SNPs tested were significant after the Bonferroni correction (*p* < 0.006) for multiple SNP testing. We did not exclude the rs4375 variant of the PLA2G6 gene from the study despite the absence of HWE, as we considered that preterm infants were not a healthy control group.

Comprehensive results are provided in Table 4.

## 3. Discussion

Ongoing research into genetic determinants may enhance our understanding of the underlying pathophysiological mechanisms and prognostic implications of patent ductus arteriosus, especially in preterm neonates who are inherently more susceptible to hemodynamic instability.

Our study focused on the analysis of polymorphisms in genes encoding enzymes and receptors involved in the prostaglandin metabolism pathway, specifically phospholipase A2, cyclooxygenase-1, prostaglandin synthetase 2, and the EP4 receptor. Genetic material was obtained from neonates born at our institution or in affiliated regional hospitals, resulting in a study cohort characterized by ethnic homogeneity. This methodological feature raises the possibility that the distribution of certain polymorphisms may differ in other European populations or globally, which may, in turn, influence the broader applicability and external validity of our findings.

Among the analyzed genes, the rs1051931 polymorphism was found to be associated with a statistically significant increase in the risk of PDA development; in neonates carrying this variant, the odds ratio (OR) for PDA occurrence was elevated by 2.49-fold. This polymorphism is located within the gene encoding phospholipase A2, an enzyme responsible for catalyzing the conversion of membrane phospholipids into arachidonic acid, a key substrate in the prostaglandin synthesis pathway. The alanine-to-valine substitution affects the gene’s active site, resulting in lower affinity for PAF-AH. Analyzing the available literature, we see that this polymorphism leads to slower degradation of the encoded protein, which translates to longer-lasting activation of proinflammatory signals.

To our knowledge, this is one of the first studies to demonstrate an association between the rs1051931 polymorphism and the incidence of patent ductus arteriosus (PDA) in neonates. The existing literature includes numerous studies investigating the relationship between polymorphisms of this gene and their involvement in the pathogenesis of cardiovascular diseases. These studies primarily focus on the gene’s role in modulating the proinflammatory response and regulating phospholipid metabolism [12,13,14].

Evidence from animal models, particularly in mice, indicates that impaired function of enzymes involved in the prostaglandin metabolism pathway—such as cyclooxygenase-1 (COX-1, *Ptgs1*) and cyclooxygenase-2 (COX-2)—is associated with a significantly increased mortality rate [15,16]. Given the high degree of conservation in biochemical pathways across mammalian species, it is plausible that similar mechanisms may be relevant in human neonates, warranting further investigation. A comparable pattern has been observed with dysfunction of the EP4 receptor, which has been linked to an increased incidence of PDA and higher neonatal mortality [17,18].

The development of hemodynamically significant patent ductus arteriosus (HsPDA) is of critical relevance in clinical practice, as disturbances in systemic perfusion associated with this condition are directly linked to an increased risk of severe complications, including intraventricular hemorrhage, necrotizing enterocolitis, and mortality. Although echocardiographic assessment remains the current gold standard for the evaluation and monitoring of ductal patency, its implementation requires access to advanced imaging equipment and highly trained personnel. Moreover, it often necessitates additional procedures in a patient population composed largely of extremely preterm and clinically fragile neonates.

In the present study, none of the investigated polymorphisms demonstrated a statistically significant association with the development of HsPDA. However, in the case of the rs4375 polymorphism, a trend toward significance was observed under the codominant, dominant, and log-additive genetic models. These preliminary findings highlight the need for further research to elucidate the potential role of this variant in the pathophysiology of HsPDA.

Our study was conducted in a population of preterm infants, including those born at extremely low gestational ages. When evaluating outcomes in this group, it is essential to consider that extremely premature neonates possess a reduced amount of smooth muscle tissue within their vasculature, which may contribute to delayed closure of the ductus arteriosus. The initial phase of ductal closure involves vasoconstriction, which is dependent on the withdrawal of prostaglandin activity.

Although certain genetic predispositions were identified, well-established clinical risk factors—such as prematurity, low birth weight, and the need for mechanical ventilation—remain the most prominent contributors to the development of PDA and HsPDA. At present, the most effective strategy for preventing PDA and its associated complications appears to be the prevention of preterm birth. In cases where premature delivery cannot be avoided, perinatal management in specialized tertiary care centers remains essential to optimize neonatal outcomes.

To our knowledge, this is the first study on the influence of gene polymorphisms on the development of PDA in premature infants. It yielded promising results, shedding new light on the development of this condition in newborns. Given the ethnic homogeneity of the study population—Caucasian—it seems reasonable to expand this study to include a larger number of children from diverse populations in the future.

## 4. Methods

### 4.1. Definitions

Patent ductus arteriosus (PDA) is defined in the literature as a persistent vascular connection between the aorta and the pulmonary artery with sustained blood flow beyond the fifth day of postnatal life. Hemodynamic significance was assessed according to the criteria established in current neonatal care guidelines (Table 5). The necessary conditions for the diagnosis of HsDPA is the assessment of the DA width, the flow in the aorta below the DA origin and the end-diastolic flow in the left pulmonary artery.

### 4.2. Diagnosis of PDA

Diagnostic assessment was conducted using echocardiography, which remains the gold standard for the detection and monitoring of the ductus arteriosus. Examinations were performed by appropriately trained medical personnel. The determination of hemodynamic significance was based on the most recent guidelines of the Polish Neonatal Society. Echocardiographic evaluations were carried out using a Samsung V8 ultrasound system equipped with a PA4-12B transducer (Samsung Medison Co., Ltd., Hongcheon, Republic of Korea).

### 4.3. Study Design and Data Collection

The study included 99 preterm infants born between 27 and 32 weeks of gestation, who were hospitalized in the Neonatal Intensive Care Unit of the Gynecology and Obstetrics Clinical Hospital (GPSK) in Poznań during 2022 and 2023.

Neonates with congenital heart defects requiring surgical intervention, as well as those who died before the fifth day of life, were excluded from the study. Medical data were collected retrospectively from clinical records documented during hospitalization. Blood samples for genetic analysis were obtained during routine blood collection performed for standard diagnostic testing.

### 4.4. Ethics

Medical data were obtained from available patient records. To minimize the need for additional procedures, blood samples for genetic testing were collected concurrently with routine blood sampling. Efforts were made to reduce the volume of blood required for genotyping to a minimum (0.5 mL). In each case, the parents were informed in advance about the planned testing, its purpose, and provided written informed consent.

To ensure confidentiality, the number of personnel involved in data processing was restricted to the minimum necessary, and all data were encrypted. The study was approved by the Bioethics Committee of the Poznan University of Medical Sciences (Resolution No. 96/22).

### 4.5. Genetic Testing Methodology

Blood for genetic polymorphism studies was collected in tubes containing EDTA (ethylene diamine tetraacetic acid). The tubes were stored at −20 °C until DNA isolation. DNA isolation from nucleated blood cells was performed using the QIAamp DNA Mini Kit (QIAGEN, Hilden, Germany), according to the manufacturer’s recommendations.

The polymorphic presented in the table were marked using polymerase chain reaction (PCR) and restriction fragment length polymorphism (RFLP). (Table 6) To select restriction enzymes specific for the SNPs we used the NEBcutter (version 3.0—accessed on 8 October 2024, http://nc2.neb.com/NEBcutter2/) (Table 7).

The separation of RFLP reaction products was carried out in a 2.5% agarose gel in 1xTBE buffer at 120 V for about 2 h in the presence of a 50 bp standard. Based on the results of electrophoretic separation after visualization in UV light, individual genotypes were determined (Figure 1).

### 4.6. Statistical Analysis

Chi-square tests, with or without Yates’ continuity correction as appropriate, were used to compare dichotomous variables. Odds ratios (ORs) with corresponding 95% confidence intervals (95% CI) were calculated to estimate the strength of associations. A *p*-value < 0.05 was considered statistically significant.

We calculated the statistical power assuming the incidence of PDA and HsDPA in preterm infants, in our hospital, during the two years of the study, which were 33.6% and 12.1%, respectively. The prevalence in our cohort was 36.4% for PSA and 21.2% for HsPDA. With a sample size of N = 99 and alpha = 0.05, the power of the test was 8.9% for PDA and 74.3% for HsPDA. The significant comparisons were further corrected (Pcorr) using Bonferroni’s correction for multiple testing (0.05/8 = 0.006).

Statistical analyses were performed using GraphPad Software (version 2024) and Statistica (version 10, 2011; StatSoft, Inc., Tulsa, OK, USA).

## 5. Conclusions

The field of research concerning genetic predispositions to specific disease entities remains in its early stages. Observations of increased disease incidence in twins and in animal models strongly suggest that certain conditions have a substantial genetic component. Our study indicates that specific polymorphisms may contribute to the development of complications such as patent ductus arteriosus, including its hemodynamically significant form. Expanding this line of research through the recruitment of larger patient cohorts—particularly those representing diverse geographic and ethnic backgrounds—may facilitate the development of a robust genomic database. Such a resource could ultimately support the identification of neonates at elevated risk and enable earlier implementation of targeted diagnostic and therapeutic strategies.

## Figures and Tables

**Figure 1 ijms-26-09274-f001:**
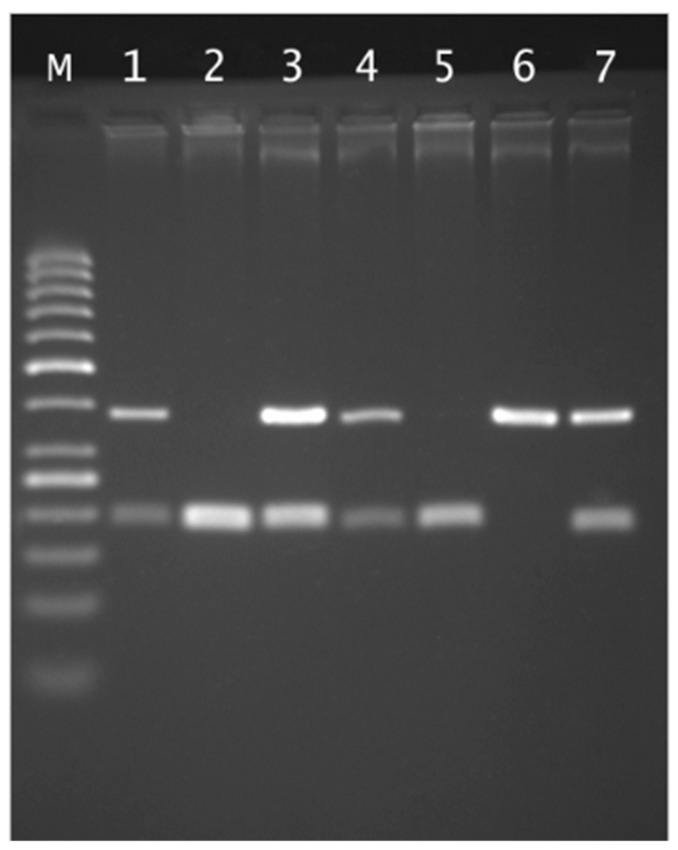
Agarose gel electrophoresis of rs4375 PLA2G6 genotyping: Lane M, 50 bp marker ladder; lane 6 CC genotype (380 bp); lanes 2 and 5 TT genotypes (199 and 181 bp); lanes 1, 3, 4 and 7 CT genotypes.

**Table 1 ijms-26-09274-t001:** The table displays the distribution of clinical variables in relation to the presence of PDA and HsPDA.

Characteristic									
	PDA		*p*	HsPDA		*p*	PDA vs. HsPDA		*p*
Sex	Female	12	*p* = 0.089	Female	7	*p* = 1.0	Female	12 vs. 7	*p* = 1.0
	Male	24		Male	14		Male	24 vs. 14	
Gestational Age (week)	<28	18	*p* = 0.111	<28	13	*p* = 0.176	<28	18 vs. 13	*p* = 0.552
	>28	18		>28	8		>28	18 vs. 8	
Birth weight (g)	<1000 g	17	*p* = 0.070	<1000 g	13	*p* = 0.080	<1000 g	17 vs. 13	*p* = 0.426
	>1000 g	19		>1000 g	8		>1000 g	19 vs. 8	
Prenatal steroid therapy	Yes	30	*p* = 0.524	Yes	18	*p* = 1.0	Yes	30 vs. 18	*p* = 1.000
	No	6		No	3		No	6 vs. 3	
Invasive ventilation	Yes	23	*p* = 0.0069	Yes	14	*p* = 0.9532	Yes	23 vs. 14	*p* = 0.552
	No	13		No	7		No	13 vs. 7	
Pharmacological ligation	Yes	21	*p* < 0.0001	Yes	21	*p* < 0.0001	Yes	21 vs. 21	*p* = 0.0017
	No	15		No	0		No	15 vs. 0	
Ibuprofen	Yes	7		Yes	7		Yes	7 vs. 7	
	No	0		No	0		No	0 vs. 0	
Paracetamol	Yes	19		Yes	18		Yes	19 vs. 18	
	No	3		No	3		No	3 vs. 3	
Complications									
NEC	Yes	11	*p* = 0.012	Yes	5	*p* = 0.501	Yes	11 vs. 5	*p* = 0.895
	No	25		No	15		No	25 vs. 15	
IVH	Yes	19	*p* = 0.0629	Yes	14	*p* = 0.1017	Yes	19 vs. 14	*p* = 0.4554
	No	17		No	7		No	17 vs. 7	
BPD	Yes	26	*p* = 0.0075	Yes	15	*p* = 0.556	Yes	26 vs. 15	*p* = 1.0000
	No	10		No	6		No	10 vs. 6	
ROP	Yes	22	*p* = 0.048	Yes	13	*p* = 1.000	Yes	22 vs. 13	*p* = 1.000
	No	14		No	8		No	14 vs. 8	

**Table 2 ijms-26-09274-t002:** Frequency of the studied single nucleotide polymorphisms (SNVs) in the entire cohort (N = 99).

Gene	rs Number	Genotypes, N (%)	Alleles	HWE (*p*-Value)
PTGS1 (COX1)	rs1236913	CC 86 (86.87%) CT 13 (13.13%) TT 0 (0.00%)	C 185 (93.43%) T 13 (6.56%)	1.000
PTGES2	rs13283456	CC 91 (91.92%) CT 7 (7.07%) TT 1 (1.01%)	C 189 (95.45%) T 9 (4.55%)	0.173
PTGER4	rs4613763	TT 68 (68.69%) TC 29 (29.29%) CC 2 (2.02%)	T 165 (83.33%) C 33 (16.67%)	1.000
PLA2G4A	rs10798059	GG 35 (35.35%) AG 52 (52.53%) AA 12 (12.12%)	G 122 (61.62%) A 76 (38.38%)	0.394
PLA2G4C	rs1549637	TT 77 (77.78%) TA 19 (19.19%) AA 3 (3.03%)	T 173 (87.37%) A 25 (12.63%)	0.177
PLA2G6	rs4375	TT 29 (29.29%) CT 60 (60.61%) CC 10 (10.10%)	T 118 (59.60%) C 80 (40.40%)	0.013
PLA2G7	rs1805017	CC 52 (52.53%) CT 40 (40.40%) TT 7 (7.07%)	C 144 (72.73%) T 54 (27.27%)	1.000
PLA2G7	rs1051931	GG 61 (61.62%) AG 33 (33.33%) AA 5 (5.05%)	G 155 (78.28%) A 43 (21.72%)	0.773

The HWE test compares observed genotype frequencies with those expected under random mating, which occurs when SNPs are not subject to selection, mutation, or genetic drift.

**Table 3 ijms-26-09274-t003:** The influence of individual SNPs on the occurrence of PDA.

Model	Genotypes	NO N (%)	YES N (%)	OR (95%CI)	*p*-Value	AIC
**rs1236913**						
Codominant	CC	54 (85.7)	32 (88.9)	1.00	0.649	133.6
	CT	9 (14.3)	4 (11.1)	0.75 (0.21–2.63)		
**rs13283456**						
Codominant	CC	57 (90.5)	34 (94.4)	1.00	1.000	134.7
	CT	5 (7.9)	2 (5.6)	0.67 (0.12–3.65)		
	TT	1 (1.6)	0 (0.0)	—		
Dominant	CC	57 (90.5)	34 (94.4)	1.00	0.474	133.3
	CT-TT	6 (9.5)	2 (5.6)	0.56 (0.11–2.93)		
Recessive	CC-CT	62 (98.4)	36 (100.0)	1.00	1.000	132.9
	TT	1 (1.6)	0 (0.0)	—		
Overdominant	CC-TT	58 (92.1)	34 (94.4)	1.00	0.651	133.6
	CT	5 (7.9)	2 (5.6)	0.68 (0.13–3.71)		
**rs4613763**						
Codominant	TT	46(73.0)	22 (61.1)	1.00	0.470	134.3
	TC	16 (25.4)	13 (36.1)	1.70 (0.70–4.14)		
	CC	1 (1.6)	1 (2.8)	2.09 (0.12–35.01)		
Dominant	TT	46 (73.0)	22 (61.1)	1.00	0.222	132.3
	TC-CC	17 (27.0)	14 (38.9)	1.72 (0.72–4.11)		
Recessive	TT-TC	62 (98.4)	35 (97.2)	1.00	0.691	133.6
	CC	1 (1.6)	1 (2.8)	1.77 (0.11–29.21)		
Overdominant	TT-CC	47 (74.6)	23 (63.9)	1.00	0.263	132.5
	TC	16 (25.4)	13 (36.1)	1.66 (0.68–4.03)		
**rs10798059**						
Codominant	GG	23 (36.5)	12 (33.3)	1.00	0.899	135.6
	GA	32 (50.8)	20 (55.6)	1.20 (0.49–2.93)		
	AA	8 (12.7)	4 (11.1)	0.96 (0.24–3.84)		
Dominant	GG	23 (36.5)	12 (33.3)	1.00	0.750	133.7
	GA-AA	40 (63.5)	24 (66.7)	1.15 (0.49–2.72)		
Recessive	GG-GA	55 (87.3)	32 (88.9)	1.00	0.815	133.7
	AA	8 (12.7)	4 (11.1)	0.86 (0.24–3.08)		
Overdominant	GG-AA	31 (49.2)	16 (44.4)	1.00	0.648	133.6
	GA	32 (50.8)	20 (55.6)	1.21 (0.53–2.76)		
**rs1549637**						
Codominant	TT	51 (81.0)	26 (72.2)	1.00	0.444	134.2
	TA	11 (17.5)	8 (22.2)	1.43 (0.51–3.98)		
	AA	1 (1.6)	2 (5.6)	3.92 (0.34–45.30)		
Dominant	TT	51 (81.0)	26 (72.2)	1.00	0.320	132.8
	TA-AA	12 (19.0)	10 (27.8)	1.63 (0.62–4.28)		
Recessive	TT-TA	62 (98.4)	34 (94.4)	1.00	0.282	132.6
	AA	1 (1.6)	2 (5.6)	3.65 (0.32–41.70)		
Overdominant	TT-AA	52 (82.5)	28 (77.8)	1.00	0.566	133.5
	TA	11 (17.5)	8 (22.2)	1.35 (0.49–3.75)		
**rs4375**						
Codominant	TT	19 (30.2)	10 (27.8)	1.00	0.950	135.7
	TC	38 (60.3)	22 (61.1)	1.10 (0.43–2.78)		
	CC	6 (9.5)	4 (11.1)	1.27 (0.29–5.56)		
Dominant	TT	19 (30.2)	10 (27.8)	1.00	0.802	133.7
	TC-CC	44 (69.8)	26 (72.2)	1.12 (0.45–2.78)		
Recessive	TT-TC	57 (90.5)	32 (88.9)	1.00	0.802	133.7
	CC	6 (9.5)	4 (11.1)	1.19 (0.31–4.52)		
Overdominant	TT-CC	25 (39.7)	14 (38.9)	1.00	0.938	133.8
	TC	38 (60.3)	22 (61.1)	1.03 (0.45–2.39)		
**rs1805017**						
Codominant	CC	32 (50.8)	20 (55.6)	1.00	0.401	134.0
	CT	25 (39.7)	15 (41.7)	0.96 (0.41–2.25)		
	TT	6 (9.5)	1 (2.8)	0.27 (0.03–2.38)		
Dominant	CC	32 (50.8)	20 (55.6)	1.00	0.648	133.6
	CT-TT	31 (49.2)	16 (44.4)	0.83 (0.36–1.88)		
Recessive	CC-CT	57 (90.5)	35 (97.2)	1.00	0.178	132.0
	TT	6 (9.5)	1 (2.8)	0.27 (0.03–2.35)		
Overdominant	CC-TT	38 (60.3)	21 (58.3)	1.00	0.847	133.7
	CT	25 (39.7)	15 (41.7)	1.09 (0.47–2.50)		
**rs1051931**						
Codominant	GG	44 (69.8)	17 (47.2)	1.00	0.028	128.7
	GA	18 (28.6)	15 (41.7)	2.16 (0.89–5.22)		
	AA	1 (1.6)	4 (11.1)	10.35 (1.08–99.38)		
Dominant	GG	44 (69.8)	17 (47.2)	1.00	0.027	128.9
	GA-AA	19 (30.2)	19 (52.8)	2.59 (1.11–6.04)		
Recessive	GG-GA	62 (98.4)	32 (88.9)	1.00	0.040	129.6
	AA	1 (1.6)	4 (11.1)	7.75 (0.83–72.25)		
Overdominant	GG-AA	45 (71.4)	21 (58.3)	1.00	0.186	132.0
	GA	18 28.6)	15 (41.7)	1.79 (0.76–4.22)		

AIC, Akaike information criteria; OR, odds ratio; 95% CI, 95% confidence interval.

**Table 4 ijms-26-09274-t004:** Association between individual SNPs and HsPDA incidence.

	Genotypes	NO N (%)	YES N (%)	OR (95%CI)	*p*-Value	AIC
**rs1236913** Codominant	CC	13 (86.7)	19 (90.5)	1.00	0.722	52.8
	CT	2 (13.3)	2 (9.5)	0.68 (0.09–5.49)		
**rs13283456** Codominant	CC	13 (86.7)	21 (100.0)	1.00	0.167	49.2
	CT	2 (13.3)	0 (0.0)	—		
**rs4613763** Codominant	TT	9 (60.0)	13 (61.9)	1.00	0.697	53.1
	TC	5 (33.3)	8 (38.1)	1.11 (0.27–4.51)		
	CC	1 (6.7)	0 (0.0)	—		
Dominant	TT	9 (60.0)	13 (61.9)	1.00	0.908	52.9
	TC-CC	6 (40.0)	8 (38.1)	0.92 (0.24–3.59)		
Recessive	TT-TC	14 (93.3)	21 (100.0)	1.00	0.417	51.1
	CC	1 (6.7)	0 (0.0)	—		
Overdominant	TT-CC	10 (66.7)	13 (61.9)	1.00	0.769	52.8
	TC	5 (33.3)	8 (38.1)	1.23 (0.31–4.93)		
**rs10798059** Codominant	GG	5 (33.3)	7 (33.3)	1.00	0.190	50.0
	GA	10 (66.7)	10 (47.6)	0.71 (0.17–3.03)		
	AA	0 (0.0)	4 (19.0)	—		
Dominant	GG	5 (33.3)	7 (33.3)	1.00	1.000	52.9
	GA-AA	10 (66.7)	14 (66.7)	1.00 (0.25–4.08)		
Recessive	GG-GA	15 (100.0)	17 (81.0)	1.00	0.125	48.2
	AA	0 (0.0)	4 (19.0)	—		
Overdominant	GG-AA	5 (33.3)	11 (52.4)	1.00	0.254	51.6
	GA	10 (66.7)	10 (47.6)	0.45 (0.12–1.79)		
**rs1549637** Codominant	TT	12 (80.0)	14 (66.7)	1.00	0.711	52.5
	TA	3 (20.0)	5 (23.8)	1.43 (0.28–7.26)		
	AA	0 (0.0)	2 (9.5)	—		
Dominant	TT	12 (80.0)	14 (66.7)	1.00	0.373	52.1
	TA-AA	3 (20.0)	7 (33.3)	2.00 (0.42–9.49)		
Recessive	TT-TA	15 (100.0)	19 (90.5)	1.00	0.500	50.7
	AA	0 (0.0)	2 (9.5)	—		
Overdominant	TT-AA	12 (80.0)	16 (76.2)	1.00	0.786	52.8
	TA	3 (20.0)	5 (23.8)	1.25 (0.25–6.29)		
**rs4375** Codominant	TT	6 (40.0)	4 (19.0)	1.00	0.350	52.8
	TC	8 (53.3)	14 (66.7)	2.62 (0.57–12.18)		
	CC	1 (6.7)	3 (14.3)	4.50 (0.34–60.15)		
Dominant	TT	6 (40.0)	4 (19.0)	1.00	0.168	51.0
	TC-CC	9 (60.0)	17 (81.0)	2.83 (0.63–12.71)		
Recessive	TT-TC	14 (93.3)	18 (85.7)	1.00	0.461	52.4
	CC	1 (6.7)	3 (14.3)	2.33 (0.22–24.92)		
Overdominant	TT-CC	7 (46.7)	7 (33.3)	1.00	0.419	52.2
	TC	8 (53.3)	14 (66.7)	1.75 (0.45–6.82)		
**rs1805017** Codominant	CC	9 (60.0)	11 (52.4)	1.00 -	0.491	52.6
	CT	5 (33.3)	10 (47.6)	1.64 (0.41–6.56)		
	TT	1 (6.7)	0 (0.0)	—		
Dominant	CC	9 (60.0)	11 (52.4)	1.00	0.650	52.7
	CT-TT	6 (40.0)	10 (47.6)	1.36 (0.36–5.22)		
Recessive	CC-CT	14 (93.3)	21 (100.0)	1.00	0.417	51.1
	TT	1 (6.7)	0 (0.0)	—		
Overdominant	CC-TT	10 (66.7)	11 (52.4)	1.00	0.389	52.2
	CT	5 (33.3)	10 (47.6)	1.82 (0.46–7.18)		
**rs1051931** Codominant	GG	7 (46.7)	10 (47.6)	1.00	0.321	52.6
	GA	5 (33.3)	10 (47.6)	1.40 (0.33–5.93)		
	AA	3 (20.0)	1 (4.8)	0.23 (0.02–2.73)		
Dominant	GG	7 (46.7)	10 (47.6)	1.00	0.955	52.9
	GA-AA	8 (53.3)	11 (52.4)	0.96 (0.26–3.63)		
Recessive	GG-GA	12 (80.0)	20 (95.2)	1.00	0.151	50.8
	AA	3 (20.0)	1 (4.8)	0.20 (0.02–2.15)		
Overdominant	GG-AA	10 (66.7)	11 (52.4)	1.00	0.389	52.2
	GA	5 (33.3)	10 (47.6)	1.82 (0.46–7.18)		

AIC, Akaike information criteria; OR, odds ratio; 95% CI, 95% confidence interval.

**Table 5 ijms-26-09274-t005:** Outlines the echocardiographic parameters employed in the definition of hemodynamically significant patent ductus arteriosus (HsPDA).

Parameter	Criterion
Ductal diameter (measured at the widest point from the pulmonary artery side)	≥1.5 mm for preterm infants below 27 weeks of gestation ≤2.5 mm for preterm infants above 27 weeks of gestation
PDA/LPA (ratio of PDA diameter to left pulmonary artery)	Small < 0.5 Moderate 0.5–1.0 Large > 1.0
La/ao (ratio of anteroposterior left atrium to aortic root)	>1.8
Left ventricular enlargement	LV end-diastolic diameter (LVDd) according to Z-score
Flow in descending aorta below the PDA origin	Absence of diastolic flow or retrograde flow during diastole
End-diastoliv flow velocity in the left pulmonary artery	>0.2 m/s
Flow assessment in the anterior cerebral artery, renal arteries and celiac trunk	Resistance index (RI) > 0.9

**Table 6 ijms-26-09274-t006:** The study included analysis of the following genetic polymorphisms.

Gene	Transcript	Variant ID	Genomic Position (GRCh38.p14)	SNP	Location
PTGS1 (COX1)	NM_000962.4	rs3842787	chr9:122371228	c.50C>T, p.Pro17Leu	Exon 2
PTGES2	NM_025072.6	rs13283456	chr9:128122474 (GRCh38.p14)	c.893G>A, p. Arg298His (isoform1)	Exon 6
PTGER4	NC_000005.10	rs4613763	chr5:40392626 (GRCh38.p14)	g.40392626T>C	Intergenic variant
PLA2G4A	NM_024420.2	rs10798059	chr1:186830478 (GRCh38.p14)	c.-70+1443G>A	Intron 1
PLA2G4C	NM_003706.2	rs1549637	chr19:48048700 (GRCh38.p14)	c.1581-312T>A	Intron 16
PLA2G6	NM_003560.2	rs4375	chr22:38143034 (GRCh38.p14)	c.609+71A>G	Intron 4
PLA2G7	NM_005084.4	rs1805017	chr6:46716485 (GRCh38.p14)	c.275G>A, p.Arg92His	Exon 4
PLA2G7	NM_005084.4	rs1051931	chr6:46705206 (GRCh38.p14)	c.1136T>A, p.Val379Glu	Exon 11

**Table 7 ijms-26-09274-t007:** The table below contains the forward and reverse primers used for the PCR of each analyzed variant.

Variant ID	Primers	Autor	Enzyme
rs3842787	5′-GGTGCCCGGTG GGGAATTTTC-3′ 5′-GAGGGGAAAGGAGGGGGTTG-3′	Yi et al., 2013 [18]	FauI
rs13283456	5′-CCCACTCTGAGGACAGAACC-3′ 5′-GAGACCCTCTTCCCTGCTG-3′	Primer3Plus [19]	EcoNI
rs4613763	5′-CAGCCATGGCATTATCACAG-3′ 5′-GATGGCTAGAATGACCCCAAT-3′	Primer3Plus [19]	BccI
rs10798059	5′-TGTGCATTTGCTCAAAGGAG-3′ 5′-ATCTTGGCTCACTGCAACCT-3′	Chang et al., 2018 [20]	BanI
rs1549637	5′-TCCACTCCCATGATCCAAACA-3′ 5′-AGAAGACCTGGAAGCCTCTA-3′	Primer3Plus [19]	BstYI
rs4375	5′-GGGGTTTATTTTGCTGGGTT-3′ 5′-CAAGGGTGATGGGGAGATC-3′	Cordeiro et al., 2010 [21]	AvrII
rs1805017	5′-TCTTCAATCACCACAGCAGC-3′ 5′-TCTGGAGAGTTTGATGGCTT-3′	Zhao et al., 2016 [22]	BclI
rs1051931	5′-ATACTGCTTTGTTCCATTGT-3′ 5′-ATCAAGATACCAAGCAAGAAC-3′	Zhao et al., 2016 [22]	SatI

## Data Availability

Due to data privacy regulations, the datasets generated and/or analyzed during the current study are not publicly accessible; however, the data presented in this study are available on request from the corresponding author.

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
