# Peer review of "Association Between Genetic Polymorphisms in the Prostaglandin Pathway and the Development of Patent Ductus Arteriosus in Preterm Infants"

_ijms, 2025, doi:10.3390/ijms26199274_

Round 1

Reviewer 1 Report

Comments and Suggestions for Authors

The submitted manuscript addresses the association between genetic polymorphisms in the prostaglandin pathway and the development of patent ductus arteriosus (PDA) in preterm neonates. The study focuses on functionally relevant variants in genes involved in arachidonic acid metabolism and prostaglandin signaling. The topic is timely and potentially impactful, as it contributes to a better understanding of the genetic basis underlying PDA susceptibility and treatment responsiveness in neonates.

While the study presents a well-defined rationale and a clearly described methodology, there are some areas that require clarification and improvement to enhance the manuscript's scientific rigor, interpretability, and translational relevance.

Major Concerns

  1. Sample Size and Statistical Power
    The study includes 99 neonates, with 36 diagnosed with PDA and 21 with HsPDA. Given the number of genotyped polymorphisms and multiple statistical models used, the risk of Type I and Type II errors is considerable. No power analysis is presented to justify the sample size. Please include a power calculation or justify how the current cohort size was determined to detect significant associations with sufficient confidence.

  2. Multiple Testing Correction
    Multiple polymorphisms were analyzed across several genetic models (codominant, dominant, recessive, overdominant). However, no correction for multiple comparisons (e.g., Bonferroni or FDR) is mentioned. Given the number of tests performed, the reported p-values—especially those close to 0.05—should be interpreted cautiously. Please address this issue and consider re-analyzing the data with appropriate adjustments.

  3. Functional Implications of rs1051931
    While the association between rs1051931 and PDA is statistically significant, the functional relevance of this variant is only briefly discussed. More detailed discussion of how this SNP might influence phospholipase A2 activity and thereby prostaglandin synthesis would strengthen the biological plausibility of your findings.

  4. Clarity on Ethnicity and Generalizability
    The manuscript mentions the ethnic homogeneity of the cohort, but the specific ethnicity is not stated. For readers to assess external validity, please clearly define the ethnic background of the study population and discuss limitations in extrapolating these findings to other populations.

    Minor Concerns and Suggestions

    1. Terminology Consistency
      The manuscript uses both “polymorphisms” and “variants” interchangeably. For clarity and consistency, choose one preferred term throughout the text, preferably “polymorphisms,” given the context of the study.

Author Response

Thank you very much for taking the time to review this manuscript. Please find the detailed responses below. I am sending you the revised manuscript. All changes are marked in red.

with respect Marcin Minta 

1. Sample Size and Statistical Power
The study includes 99 neonates, with 36 diagnosed with PDA and 21 with HsPDA. Given the number of genotyped polymorphisms and multiple statistical models used, the risk of Type I and Type II errors is considerable. No power analysis is presented to justify the sample size. Please include a power calculation or justify how the current cohort size was determined to detect significant associations with sufficient confidence.

  1. Response - Agree. I have modified and supplemented the manuscript as suggested. The changes are in red.

2. Multiple Testing Correction
Multiple polymorphisms were analyzed across several genetic models (codominant, dominant, recessive, overdominant). However, no correction for multiple comparisons (e.g., Bonferroni or FDR) is mentioned. Given the number of tests performed, the reported p-values—especially those close to 0.05—should be interpreted cautiously. Please address this issue and consider re-analyzing the data with appropriate adjustments.

      2. Response - Agree. I have modified and supplemented the manuscript as suggested. The changes are in red.

3. Functional Implications of rs1051931
While the association between rs1051931 and PDA is statistically significant, the functional relevance of this variant is only briefly discussed. More detailed discussion of how this SNP might influence phospholipase A2 activity and thereby prostaglandin synthesis would strengthen the biological plausibility of your findings.

     3. Response - Agree. I have modified and supplemented the manuscript as suggested. The changes are in red.

4. Clarity on Ethnicity and Generalizability
The manuscript mentions the ethnic homogeneity of the cohort, but the specific ethnicity is not stated. For readers to assess external validity, please clearly define the ethnic background of the study population and discuss limitations in extrapolating these findings to other populations.

     4. Response - I have modified and supplemented the manuscript as suggested. The changes are in red.

5. Terminology Consistency
The manuscript uses both “polymorphisms” and “variants” interchangeably. For clarity and consistency, choose one preferred term throughout the text, preferably “polymorphisms,” given the context of the study.

     5. Response - Agree. I have,modified rÄ™kopis zgodnie z sugestiÄ….  

Reviewer 2 Report

Comments and Suggestions for Authors

In their article titled "Association Between Genetic Polymorphisms in the Prostaglandin Pathway and the Development of Patent Ductus Arteriosus in Preterm Infants," Minta et al. have described their study in which they assessed whether certain variants in genes that code for proteins of the prostaglandin metabolic pathway are associated with a higher risk of PDA in preterm infants. They have found a significant association for the rs1051931 variant in the PLA2G7 gene. While their work does present a contribution to the current field, I have several comments that should be addressed prior to publication:

1) For each variant, please also compare the allelic frequencies between (Hs)PDA and controls, and calculate the OR, 95% CI, and p-values. This is usually done together with the genotype models and can be added to Tables 5 and 6. If any significance is found, it should be commented on in the text of the results.

2) Please add a table containing the forward and reverse primers used for the PCR of each analyzed variant.

3) Please add an original image of genotype visualization after gel electrophoresis as a figure (1 is enough)

4) Which tools were used to select restriction enzymes specific for the SNPs? This should be added to the methods section.

5) Table 2: Be careful with the spelling of the headers; they seem to be in Polish? I would rename "variant number" to "variant ID" and "allele" to "SNP." I also suggest adding additional information about the variants to the table, including the transcript (after the gene), genomic position, sequence change according to HGVS, and location of the SNP within the gene. For instance:

Gene Transcript Variant ID Genomic position SNP Location Enzyme
PLA2G4A NM_024420.3 rs10798059 chr1:186830478 c.-70+1443G>A, p.? Intron 1 BanI
PTGS1 NM_000962.4 rs1236913 chr9:122371200 c.22T>C, p.(Trp8Arg) Exon 2 FauI

6) Line 103-105: Please add details about how hemodynamic significance was determined based on the criteria in Table 1. How many criteria had to be fulfilled?

7) Under Table 4, you should define HWE. Also, you should comment in the results section that the rs4375 variant in the PLA2G6 gene is not in HWE (p-value 0.013).

8) Line 168: You should add specific details about the variant associated with PDA, including the model and OR: for the dominant model the OR was 2.59 and for the homozygous model (AA vs GG) the OR was 10.35. Please note that the heterozygous model (GA vs AA) and recessive models, while having a p<0.05, should not be considered significant since the 95% CI for OR includes 1, i.e., no increased risk. Actually, lines 196-21 should be moved to the results section.

9) Line 170 Table 4 should be Table 5, and line 178 Table 5 should be Table 6.

10) You should add a paragraph to the end of the discussion section highlighting the main limitations and strengths of your study

Author Response

Thank you very much for taking the time to review this manuscript. Please find the detailed responses below. All modifications to the manuscript have been made in red. 

with respect, Marcin Minta

1) For each variant, please also compare the allelic frequencies between (Hs)PDA and controls, and calculate the OR, 95% CI, and p-values. This is usually done together with the genotype models and can be added to Tables 5 and 6. If any significance is found, it should be commented on in the text of the results.

1) Response - Agree. I have modified the manuscript as suggested. The changes have been made in red.

2) Please add a table containing the forward and reverse primers used for the PCR of each analyzed variant.

2) Response - The appropriate table has been added to the manuscript.

3) Please add an original image of genotype visualization after gel electrophoresis as a figure (1 is enough)

3) Response - The appropriate figure has been added to the manuscript.

4) Which tools were used to select restriction enzymes specific for the SNPs? This should be added to the methods section.

4) Response - The missing information has been added to the manuscript.

5) Table 2: Be careful with the spelling of the headers; they seem to be in Polish? I would rename "variant number" to "variant ID" and "allele" to "SNP." I also suggest adding additional information about the variants to the table, including the transcript (after the gene), genomic position, sequence change according to HGVS, and location of the SNP within the gene.

5) Response - Agree. I have modified the manuscript as suggested. Errors have been corrected. The missing information has been added to the manuscript. The changes are in red.

6) Line 103-105: Please add details about how hemodynamic significance was determined based on the criteria in Table 1. How many criteria had to be fulfilled?

6) Response - Agree. I have modified and supplemented the manuscript as suggested. The changes are in red.

7) Under Table 4, you should define HWE. Also, you should comment in the results section that the rs4375 variant in the PLA2G6 gene is not in HWE (p-value 0.013).

7) Response - Agree. I have modified the manuscript as suggested. The changes are in red.

8) Line 168: You should add specific details about the variant associated with PDA, including the model and OR: for the dominant model the OR was 2.59 and for the homozygous model (AA vs GG) the OR was 10.35. Please note that the heterozygous model (GA vs AA) and recessive models, while having a p<0.05, should not be considered significant since the 95% CI for OR includes 1, i.e., no increased risk. Actually, lines 196-21 should be moved to the results section.

8) Response - Agree. I have modified and supplemented the manuscript as suggested. The changes are in red.

9) Line 170 Table 4 should be Table 5, and line 178 Table 5 should be Table 6.

9) Response - I agree. The incorrect entries have been corrected.

10) You should add a paragraph to the end of the discussion section highlighting the main limitations and strengths of your study

10) Response - I agree. I have provided the missing information.
